# Ferrocene-Doped Polystyrene Nanoenzyme and DNAzyme Cocatalytic SERS Quantitative Assay of Ultratrace Pb^2+^

**DOI:** 10.3390/nano12081243

**Published:** 2022-04-07

**Authors:** Chongning Li, Zhenghong Wang, Zhiliang Jiang

**Affiliations:** 1School of Public Health, Guilin Medical University, Guilin 541199, China; lcn7882342@163.com (C.L.); zh27685@163.com (Z.W.); 2Guangxi Key Laboratory of Environmental Pollution Control Theory and Technology, Guilin 541006, China

**Keywords:** Pb^2+^, aptamer, ferrocene doped polystyrene nanoenzyme, cocatalysis, SERS

## Abstract

A new, stable and high-catalytic activity ferrocene-doped polystyrene nanosphere (PN_Fer_) sol was prepared by the hydrogel procedure and characterized by electron microscopy and molecular spectroscopy. Results show that the nanosol exhibits excellent catalysis of the new indicator nanoreaction between AgNO_3_ and sodium formate to generate nanosilver with strong surface-enhanced Raman scattering (SERS), resonance Rayleigh scattering (RRS) and surface plasmon resonance absorption (Abs) trimode molecular spectral signals. This new nanocatalytic amplification trimode indicator reaction was coupled with the G-quadruplex DNAzyme catalytic amplification of Pb^2+^ aptamer to fabricate a new SERS quantitative/RRS/Abs assay platform for the determination of ultratrace amounts of Pb^2+^. The Pb^2+^ content in water samples was analyzed with satisfactory results.

## 1. Introduction

Lead (Pb) is a toxic heavy metal; it has caused great concern due to its damage to the kidneys and nervous system. The measurement of trace Pb^2+^ is becoming more and more common and urgent in the food safety, environmental, clinical and toxicology fields. At present, methods for the determination of Pb include dithizone colorimetry, flame atomic absorption spectrometry, graphite furnace atomic absorption spectrometry, hydride generation atomic fluorescence spectrometry, catalytic polarography and so on [1,2]. The atomic spectral methods use large-scale instruments with high costs. Classical dithizone colorimetry demonstrates low sensitivity and some toxic reagents are used, which poses a threat to the health of inspectors. Recently, some new methods for Pb have been attractive to analysts. One of the most important methods is the aptamer (Apt) assay, which is simple with good selectivity. Based on the interaction between single-stranded DNA (ssDNA) and nanogold (AuNPs), a turn-on plasmonic absorption assay for the detection of 8.0 nM Pb^2+^ was fabricated [3], and the sensitivity was not high. Acoupled DNAzyme-activated hybridization chain reaction with bio-barcodes and a surface-enhanced Raman scattering (SERS) were developed for assay of Pb^2+^ with a dynamic range of 1.0 × 10^–13^ M to 1.0 × 10^−7^ M [4]. However, the SERS signal was not linear to the concentration. In addition, the process was complicated. Thus, it is necessary to develop a simple and sensitive Apt method for Pb^2+^-based DNAzyme and nanozyme catalytic amplification. To our best knowledge, there are no reports about the preparation of a stable and high-catalytic activity ferrocene (Fer)-doped polystyrene nanosphere (PN_Fer_) sol, the PN_Fer_-AgNO_3_- sodium formate (SF) nanosilver indicator reaction, or the coupled DNAzyme catalytic reaction to co-amplify the SERS/resonance Rayleigh scattering (RRS)/absorption (Abs) trimode signals for determination of ultratrace Pb^2+^.

DNA enzymes (DNase) were synthesized simply by Apts, with the advantages of specificity, low costs and good stability. The method has been used in analytical chemistry, such as in metal ion and protein detection [5,6,7,8,9]. Zhang et al. [10] proposed a lead ion detection method without labeling reported molecules. Pb^2+^ induces chemically modified DNase to cleave unlabeled substrates to amplify the response signal, with a detection limit of 0.12 nM Pb^2+^ and linear range of 0.1–30 nM. SERS technology can provide a wealth of structural information, and its sensitivity can even reach the single-molecule level. It is a powerful tool for trace bioanalysis and material characterization [11,12,13]. Although some metal ions such as Pb(II) exhibit no SERS signal, clever strategies can be used to detect them, such as complexes and SERS probes [14,15,16]. Due to their simplicity, multiple options and sensitivity, bimodal molecular probes are more widely used than the single-mode sensor. They include two-mode analysis of fluorescence/colorimetry, fluorescence/light scattering, fluorescence/SERS etc. They can display multiple output signals to realize double detection of the analyte to be tested [17,18,19]. Recently, Li et al. [20] developed a SERS/RRS dual mode aptamer assay platform for quantification of trace pollutants such as urea, based on single-atom Fe-doped carbon dot nanocatalytic amplification.

Polystyrene is a stable and common organic polymer which has been used in the field of immunoassay as a microsphere carrier. Recently, there has been interest in utilizing a combination of polystyrene nanoparticles for spectral and electrical analysis, such as the detection of trace amounts of H_2_O_2_, dithiothreitol and Pb(II) [21,22,23]. Ferrocene (Fer) has a sandwich structure with d–π interactions between the Fe(II) center and the cyclopentadienyl moieties, which makes it unique in terms of its chemical properties such as being an electron-rich molecule, which is often used to promote electron transfer; this property allows it to be utilized in chemical sensors, including fluorescent and electrochemical sensors for the detection of Pb^2+^ [24,25,26]. Xu et al. prepared Fer–triazole derivatives for multisignaling detection of Cu^2+^ [27]. In this paper, a soap-free emulsion polymerization procedure was used to prepare ferrocene-doped polystyrene nanosphere (PN_Fer_), and its nanocatalytic reaction was coupled with the DNAzyme catalytic reaction to construct a trimode method for ultratrace Pb^2+^.

## 2. Materials and Methods

### 2.1. Instruments and Reagents

Resonance Rayleigh scattering spectra were scanned by a Hitachi F-7000 FL spectrophotometer (Hitachi High-Tech Co., Ltd., Tokyo, Japan). The absorption spectrum was measured with a TU-1901 dual-beam ultraviolet-visible spectrophotometer (Beijing Puxi General Instrument Co., Ltd. Company, Beijing, China). A DXR smart Raman spectrometer (Thermo Company, Waltham, MA, USA) was used to obtain SERS spectra with a laser wavelength of 633 nm, a laser power of 3.0 mW, a slit of 25 µm, and an acquisition time of 5 s. A 16K desktop centrifuge (Zhuhai Dark Horse Medical Instrument Co., Ltd., Zhuhai, China); KQ3200DB CNC ultrasonic cleaner (Kunshan Ultrasonic Instrument Co., Ltd., Nanjing, China, ultrasonic electric power 150 w, working frequency 40 KHz); HH-2 electric heating constant-temperature water bath (Shanghai Weicheng Instrument Co., Ltd., Shanghai, China); SYZ-550-type quartz sub-boiling water distiller (Jiangsu Jingbo Instrument Factory, Nanjing, China); DHG-9023A electric heating constant-temperature blast drying oven (Shanghai Jinghong Experimental Equipment Co., Ltd., Shanghai, China); FB224 automatic internal school electronic analytical balance (Shanghai Sunny Hengping Scientific Instrument Co., Ltd., Shanghai, China); DF-101S type heat-collecting constant-temperature heating magnetic stirrer (Gongyi Yuhua Instrument Co., Ltd., Zhengzhou, China); GL-25MS type low-temperature high-speed centrifuge (Shanghai Luxiangyi Centrifuge Instrument Co., Ltd., Shanghai, China); Nanoparticle size and Zeta potential analyzer (Malvern Co., Malvern, UK); S-4800 field emission scanning electron microscope (Hitachi High-Tech, Tokyo, Japan) and Talos F200S transmission electron microscope (Thermo Fisher Scientific China Co., Ltd., Shanghai, China) were used.

A Pb^2+^ aptamer with sequence of 5’-3’ GGT TGG TGT GGT TGG (Sangong Bioengineering Co., Ltd., Shanghai, China); Styrene (CP, Guangxi Long Science Co., Ltd., Nanning, China); Chloroauric acid (AR, Shanghai Mike Lin Biochemical Co., Ltd., Shanghai, China); Potassium persulfate (GR, Tianjin Komiou Chemical Reagent Co., Ltd., Tianjin, China); Sodium lauryl sulfate (SDS, Guangzhou Medical Station Chemical Reagent Company, Guangzhou, China); Sodium bicarbonate (AR, Guangxi Donglong Chemical Co., Ltd., Nanning, China); pH 4.4 NaAc-HAc buffer solution mixed 3.7 mL 0.2 mol/L NaAc solution and 6.3 mL 0.2mol/L HAc solution to a 10 mL; 1.00 mM Pb^2+^ (PbNO_3_); 1 M AgNO_3_; 0.1 mM acetic acid; 0.1 M sodium acetate; 1 M sodium formate (SF); Ferrocene (Fer) (AR, Tianjin Zhiyuan Chemical Reagent Co., Ltd., Tianjin, China); 0.1 μM heme (HM) (Tianjin Zhiyuan Chemical Reagent Co., Ltd., Tianjin, China); 10^−5^ M Vitoria blue B (VBB); 10^−5^ M Victoria blue 4R (VB4R) (Tianjin Zhiyuan Chemical Reagent Co., Ltd., Tianjin, China); and 10^−5^ M rhodamine 6G (Rh6G) (Tianjin Zhiyuan Chemical Reagent Co., Ltd., Tianjin, China) were used. For the preparation of the CO-saturated solution, 20 mL of deionized water was added to a 30 mL gas collection bottle and placed into a 25 °C water bath. Under standard atmospheric pressure, CO gas was poured into it for 10 min to prepare 0.93 mM CO saturated solution. For preparation of the Tris-HCl buffer solution, 510 μL of 0.1 M HCl solution was added to 500 μL of 0.1 M Tris solution. For preparation of the Fer solution, 100 mg of Fer and 0.1 g of sodium lauryl sulfate (SDS) were added to a 200 mL three-necked flask, then 100 mL of water was added, the solution was stirred at constant temperature for 1 h, and the concentration according to the added Fer was calculated as 5.37 mM, diluted step-by-step. For the preparation of oxide of 3,3’,5,5’-tetramethylbenzidine (TMBox), 100 mg 3,3’,5,5’-tetramethylbenzidine (TMB) (AR, Tianjin Zhiyuan Chemical Reagent Co., Ltd., Tianjin, China) was weighed in a 250 mL Erlenmeyer flask under ultrasonic conditions, 35 mL absolute ethanol, 8.3 mL 0.01 M NaOH, 18.9 mL 1 M H_2_O_2_ and 18.2 mL 0.1 M Tris-HCl were added. After 45 min in a 45 °C water bath, the solution turned dark blue. Then, the solution was freeze-dried to obtain a TMBox solid. Next, 0.0031 g TMBox was dissolved in 5 mL secondary water to obtain 0.62 mg/mL TMBox. In the experiment, it was diluted and used step by step. The reagents used were analytically pure, and the experimental water was boiled twice before use.

**Preparation of polystyrene nanospheres:** Preparation of polystyrene nanospheres (PN) was as follows, 5, 10, 15, and 20 mL styrene, respectively, were added into a 500 mL three-necked flask containing 20 mL water, and passed through nitrogen to exclude air. Following this, 0.1 g SDS was added and heated at 70 °C for 1 h under stirring. Next, 0.1 g potassium persulfate and 0.1 g NaHCO_3_ were added, diluted to 100 mL with water, reacted for 10 h to complete the reaction, and let cool to room temperature. The obtained white emulsion was centrifuged at a low speed to remove the unreacted styrene on the upper layer, dialyzed with a 500 Da dialysis membrane for 12 h to remove inorganic salts, and finally the obtained polystyrene nanosphere (PN) solution was vacuum-dried to a constant weight. A white powdery PN was obtained, which was numbered PN_1-4_ according to the quality of the styrene. Finally, 0.10 g PN_1-4_ was dissolved in 100 mL water to prepare a 1 g/L solution, and diluted step by step during use.

**Preparation of ferrocene-doped polystyrene nanozyme:** For PN_Fer1-3_ preparation, 20, 50, 100 mg Fer, 15 mL styrene, and 0.1 g SDS were added, respectively, into a 500 mL three-necked flask containing 20 mL deionized water, and passed through nitrogen to remove air. It was heated at 70 °C for 1 h under stirring. Next, 0.1 g potassium persulfate and 0.1 g NaHCO_3_ were added, diluted to 100 mL with water, reacted for 10 h to complete the reaction, and let cool to room temperature. The obtained white emulsion was centrifuged at a low speed to remove the unreacted styrene on the upper layer, dialyzed with a 500 Da dialysis membrane for 12 h to remove inorganic salts, and finally the obtained ferrocene-doped polystyrene nanosphere (PN_Fer_) solution was dried under vacuum to constant weight. A light yellow powdered PN was obtained, which was numbered PN_Fer1_, PN_Fer2_, or PN_Fer3_ according to the different mass ratio of ferrocene to styrene. Finally, 0.10 g PN_Fer_ was dissolved in 100 mL water to prepare a 1 g/L solution, and diluted step by step during use.

### 2.2. Procedure for RRS and SERS Detection of Pb^2+^

In a 5 mL graduated tube with stopper, 100 µL 100 mg/L polystyrene nanoparticle solution, 150 µL pH 4.4 NaAc-HAc buffer (88.8 mM in NaAc concentration), 120 µL 0.1 µM Apt, 90 µL 0.1 µM heme and a certain concentration of Pb^2+^ solution were added, then mixed well. After 15 min, 100 µL 0.01 M AgNO_3_ and 80 µL 1.0 M formate were added, diluted to 1.5 mL and shaken well, placed in a water bath at 85°C for 10 min, and cooled with ice water to terminate the reaction. The solution was placed in a quartz four-way cuvette, at 350 V, excited slit = emission slit = 5 nm, and a fluorescence spectrophotometer was used to record the RRS signal at 370 nm (I_370nm_). With no Pb^2+^ as a blank, the RRS peak at 370 nm (I_370nm_)_0_ was measured and ΔI = I_370nm_ − (I_370nm_)_0_ was calculated. Then 50 μL of 10 μM VBB was added to the reaction solution, the Raman signal recorded at 1618 cm^−1^ under the power of 2.0 mW and a slit of 25.0 nm, and the value of ΔI_1618 cm^−1^_ = I_1618 cm^−1^_ − (I_1618 cm^−1^_)_0_ obtained.

## 3. Results and Discussion

### 3.1. Analysis Principle

At 70 °C, potassium persulfate was used as the initiator and sodium lauryl sulfate sodium as the surfactant to initiate the polymerization reaction of styrene to form polystyrene nanospheres (PN). When Fer was added, the PN_Fer_ was obtained (Figure 1). Under room temperature, AgNO_3_-SF is unreactive. When nanospheres were added as a catalyst, the reaction can be completed within a few minutes. When Apt was added, because Apt has a better affinity with the nanospheres, Apt will coat the surface of the nanospheres to reduce its catalytic activity, and will therefore produce low SERS/RRS/Abs signal. When Pb^2+^ was added, Apt will specifically bind to Pb^2+^, and at the same time, it will combine with the existing heme (HM) to form DNase. Both DNase and the released nanospheres have a catalytic effect on the AgNO_3_-SF indicator reaction to enhance the trimode signals of SERS/RRS/Abs. Thus, a highly sensitive, low-cost trimode assay platform was proposed for the detection of Pb^2+^.

### 3.2. Characterization of PN_Fer_ and Nanocatalytic Analysis System

Figure 2a–d are electron microscope images of PN_3_, PN_Fer3_, PN_Fer3_-Apt-Pb^2+^-AgNO_3_-SF and PN_Fer3_-Apt-AgNO_3_-SF systems. It can be seen that the PN_3_ average particle size was larger than that of PN_Fer3_, indicating that after PN doping with the electron-rich molecule Fer, the particle size became small and the particle size distribution was uniform. This may be related to the increase in electron density and intermolecular forces. The energy spectrum (Figure 2b) shows that there are three peaks at 0.46, 6.4 and 7.1 keV, ascribed to the Fe element of Fer. There are less spherical AgNPs in the blank system due to the inhibition of Apt, and the average size is 42 nm (Figure 2c). Upon addition of Pb^2+^, the Apt reaction turned on the catalytic AgNP nanoreaction to form a large number of AgNPs with an average size of 25 nm (Figure 2d). Both energy spectra demonstrated that there was a peak at 3.0 keV ascribed to the Ag element (Figure 2c,d). 

**Molecular spectra and X-ray diffraction (XRD)****characterization of PN_3_****and PN_Fer3_**_:_ RRS is a simple and sensitive technique for studying nanoparticles, and was chosen for use. Figure 3a,b show that the 20 mg/L PN_3_ and 20 mg/L PN_Fer3_ exhibit the biggest RRS peak at 320 nm and 360 nm respectively. As the PN_3_ and PN_Fer3_ concentration increases, the RRS signal increases linearly, due to the particle numbers increased. Figure 3c shows that there are no UV absorption peaks for both materials. Figure 3d shows the emission spectra of nanomaterials at an excited wavelength of 250 nm, none of the both shows any fluorescence peak, the two peaks are ascribed to Rayleigh scattering. The SERS spectra of PN_3_ and PN_Fer3_ materials were studied, and no SERS signal appeared. The nanomaterial was prepared according to the experimental method, was vacuum-freeze-dried to obtain the solid samples, and was recorded with a Fourier transform infrared spectrometer. Figure 4a shows the infrared spectra of PN_3_ and PN_Fer3_. Among them, 760 cm^−1^ is the vibration peak caused by CH that is not in the plane of styrene, and the stretching vibration peaks caused by CH at 2859 and 2926 cm^−1^ are 3023 cm^−1^. The nearby characteristic peaks were caused by the =CH stretching vibration. At the same time, the medium-strong peak at 1598 cm^−1^ indicates that =CH exists in aromatic substances. The two characteristic peaks at 1450 and 1495 cm^−1^ directly show that the material was synthesized from styrene, and the stretching vibration peak of C=C does not appear at 1640 cm^−1^, indicating that styrene has polymerized to form polystyrene. The characteristic peak at 3075 cm^−1^ was the stretching vibration peak caused by CH in Fer, and the characteristic peak at 1101 cm^−1^ was caused by the σ bond formed by the metal ion and the phenyl compound, indicating that the doped Fer has a certain effect. Figure 4b shows that in the range of 2θ = 15–25°, there are a wide range of structural belts composed of disordered carbon elements in the two materials. The FeC structure formed by Fer and polystyrene appeared in the range of 2θ = 31° and 44° in PN_Fer3_, indicating that the doping of Fer was successful.

### 3.3. SERS, RRS and Abs Spectra of Nanocatalysis System

The effects of the four probes (VBB, VB4R, TMB_OX_, and Rh6G) on the system were studied (Appendix A–d). With an increased Pb^2+^ concentration, the SERS peaks increased due to the formation of more SERS active AgNPs. It can be seen that the system has higher signal intensity, less interference of spurious peaks and better sensitivity upon addition of VBB, so the VBB molecular probe was selected for use. The Pb^2+^-Apt-heme-PN_3_-AgNO_3_-formate-VBB system shows strong peak changes at the Raman shifts of 1167, 1202, 1398, and 1614 cm^−1^. The SERS effect at 1617 cm^−1^ was ascribing to the vibration of C=C in the benzene ring frame. Appendix A shows that PN_3_ has a good catalytic performance for the AgNO_3_-SF reaction, and the SERS signal was positively correlated with the PN_3_ concentration. Appendix A shows that with the increase in Apt, its inhibitory effect in the Apt-PN_3_-AgNO_3_-formate system gradually decreases due to the formation of fewer AgNPs. Appendix A shows that with the increase in heme content, the catalytic capacity of the DNase produced increases, but the catalytic performance was limited. Figure 5a shows that as regards the heme-Apt-Pb^2+^-PN_3_-AgNO_3_-formate system, under the action of two enzymes, the reaction system has a good detection ability for Pb^2+^. In the comparison of the catalytic performance of the heme-Apt-Pb^2+^-AgNO_3_-formate detection system (Figure 5b,c), the PN_Fer3_ signal was the strongest and was chosen for the assay. 

Results (Appendix A) show that after adding PN_3_, the RRS signal of the system was much higher than the blank without PN_3_, which proves that PN_3_ has a catalytic effect on the AgNO_3_-formate system. The RRS signal has a positive correlation with the PN_3_ concentration. After selecting the appropriate PN_3_ concentration and adding Apt (Appendix A), because Apt coats on the surface of PN_3_, its catalytic performance was reduced. Therefore, as Apt increases, the lower the RRS signal produced due to the formation of fewer nanosilvers. When Pb^2+^ was added, the Pb^2+^-Apt complex detaches from the surface of PN_3_ and combines with heme to form a DNase. With the increase in heme, DNase increases, the catalytic ability becomes stronger, and the RRS signal increases linearly (Appendix A). Figure 6a shows that when Pb^2+^ increases, the Pb^2+^-Apt complex detaches from the PN_3_ surface, releases more PN_3_, and the catalysis is enhanced. The more DNase formed, the stronger the catalytic effect on the system, and the RRS signal increases linearly with the concentration of Pb^2+^. Figure 6b,c are the RRS spectra of PN_3_/PN_Fer3_ and DNase for the detection of Pb^2+^. In contrast, the PN_Fer3_ was the most sensitive, and was chosen for use.

Appendix A shows that the absorption peak at 430 nm increased with the PN_3_ catalyst concentration increasing, and the yellow gradually deepened. The corresponding UV absorption signal also increases linearly with the increase in PN_3_ concentration due to the formation of more nanosilvers, which also shows that PN_3_ material has excellent catalytic ability. The peak was ascribed to the surface plasmon resonance absorption peak of nanosilvers. Appendix A shows that the acetic acid–sodium acetate buffer with ph = 4.4 is the best condition. Appendix A shows that as the Apt increases, the more PN_3_ is coated by Apt and the stronger the inhibition ability for the AgNO_3_-formate reaction, and it shows that the UV absorption signal decreases linearly with the concentration of Apt. Appendix A shows that as the concentration of HM increases, the DNase produced gradually increases and the catalytic ability increases. As the concentration of Pb^2+^ increases, the released PN_3_ increases, the DNase generated gradually increases, and the catalytic ability increases due to the formation of more nanosilvers (Figure 7a). Figure 7b,c show that PN_Fer3_ has the strongest Abs signal in the determination of Pb^2+^ and has a higher correlation. Comparing the linear equations, it is found that the PN_Fer3_ has the best catalytic performance, and the result is the same as the RRS result. 

### 3.4. Catalytic Reaction Mechanism of Nanoenzymes and DNase

Formate and silver nitrate can generate porous silver [28], but this reaction requires a microwave reaction vessel to keep pressure and temperature. Under the experimental conditions, it is difficult for the reaction between AgNO_3_ and SF to take place. After adding the catalyst, a fast reaction can be realized, AgNO_3_ is reduced to AgNPs, and SF is oxidized to CO_2_. Appendix A shows a comparison of the linear equations for catalysis of several nanomaterials. It can be seen that PN_Fer3_ has the strongest catalytic performance, with the highest slope. The catalysis of PN_Fer3_ with slope of 236.6 was larger than the sum (109.7 + 9.81 = 119.51) of PN_3_ (109.7) and Fer (9.81), in the SERS method. This was related to the coupling of PN_3_ and Fer electrons.

Figure 8 shows a schematic diagram of the catalytic mechanism of PN_Fer3_ to enhance the signal. In the absence of nanocatalyst, the nanoreaction speed is slow, the product is small, and the RRS/SERS/Abs signal is low. PNs have a small particle size, large specific surface area, and can be stably and uniformly dispersed in aqueous solution. The surface of PNs provides reaction sites for the AgNO_3_ and SF reaction system. Fer is rich in electrons and has good thermal and chemical stability [29]. Due to poor water solubility, it is rarely used in aqueous solution. The use of polystyrene nanospheres doped with ferrocene can effectively solve this problem. The excellent electronic conductivity of ferrocene enhances the catalytic performance of PNs. After optimizing the proportion of doped-ferrocene, high-efficiency catalytic performance PN_Fer3_ can be achieved. The main reason is that the nanosurface electrons and the electrons of Fer were coupled organically to enhance the electron density. Those electrons accelerate Ag^+^-formate redox electron transfer to enhance the nanoreaction.

### 3.5. Conditional Selection

As the concentration of PN_Fer3_ increased, the RRS signal gradually increased due to the formation of more nanosilvers (Appendix A). When the concentration of PN_Fer3_ exceeded 6.67 mg/L, the RRS signal of the solution decreased sharply due to nanosilver aggregations. Therefore, the concentration of 6.67 mg/L PN_3_ in this experiment was chosen. When the NaAc-HAc buffer concentration calculated with NaAc is 8.88 mM (pH = 4.4), the RRS signal change value of the system is the largest, so the NaAc-HAc buffer condition was selected (Appendix A). As the concentration of Apt increases, the system produced a stronger RRS signal (Appendix A). However, when the Apt concentration was greater than 8 nM, too much Apt will encapsulate PN_Fer3_, causing the amount of catalyst in the system to decrease, and the RRS signal will decrease. So, 8 nM Apt was selected in this experiment. For the same reason, when the concentration of HM is low, it has a positive effect on the system, and when its concentration is greater than 6 nM, it will affect the catalytic performance and cause the signal to decrease (Appendix A). Therefore, the concentration of HM was selected as 6 nM. As shown in Appendix A, the reaction temperature should be maintained above 75 °C. If the temperature is too low, the reaction cannot be started. If the temperature is too high, the enhanced blank that caused the analytical signal will decrease. At 85 °C, the analytical signal is the strongest. A reaction time of 15 min was optimal (Appendix A). The above conditions were selected for use. 

### 3.6. Work Curve

Under the chosen experimental conditions, the working curve was plotted with different concentrations of the analyte and the corresponding signal intensity changes. The analysis characteristics are shown in Table 1. Among the working curves for Pb^2+^ determination, the PN_Fer3_ SERS analytical system was more sensitive than the RRS and Abs, with a detection limit (DL) of 0.03 nM. In addition, the linear range was widest. Compared with the RRS and Abs methods, the blank of the SERS system was very low due to the large AgNPs with low SERS activity. Thus, the PN_Fer3_ catalytic SERS analysis system was used to measure Pb^2+^. Obviously, the two methods were simpler than the SERS, without molecular probes of VBB. Of course, the Abs method had the lowest cost. Compared with the reported methods (Appendix A) [30,31,32,33,34,35,36,37,38], this method is one of the more sensitive methods. 

### 3.7. Influence of Coexisting Substances

According to the experimental method, the relative error was within ±10%, and the influence of coexisting substances on the detection of 1 nM Pb^2+^ (Appendix A) was investigated. The results show that some common ions have little effect on the determination of Pb^2+^, indicating that this method has better selectivity.

### 3.8. Sample Analysis

The samples, two of farmland water, two of lake water, and one of industrial wastewater, were taken with sampling bottles. Each 10 mL water sample was filtered with filter paper to remove suspended solids, then centrifuged at 12,000 rpm/g for 10 min, and finally filtered with a 0.45 μm microporous filter membrane. It was diluted to obtain the sample solution. The SERS method of the PN_Fer3_-Pb^2+^-Apt-HM-AgNO_3_-SF system was used to determine Pb^2+^ content five times. Then, a certain amount of Pb^2+^ was added to the sample to obtain the recovery (Table 2). These results were agreement with that of atomic absorption spectrometry, the relative standard deviation (RSD) was between 3.7% and 7.8%, and the recovery rate was between 91.2% and 108.7%. Those indicate that the SERS assay was accurate and reliable.

## 4. Conclusions

A stable and high-catalytic activity ferrocene-doped polystyrene nanosphere sol catalyst was prepared. It was found that it can effectively catalyze the AgNO_3_-SF reaction to generate nanosilver with strong SERS/RRS/Abs trimode signals. The Apt of analyte lead ion was added to inhibit the catalytic ability. When the analyte lead ion exists, Apt combined with lead ions to generate G-quadruplex DNA enzyme in the presence of heme, and released ferrocene-doped polystyrene nanospheres. This DNA enzyme can also catalyze the AgNO_3_-SF reaction to generate nanosilver. Thus, a new di-enzyme trimode analysis method was established for detecting trace lead ions. 

## Figures and Tables

**Figure 1 nanomaterials-12-01243-f001:**
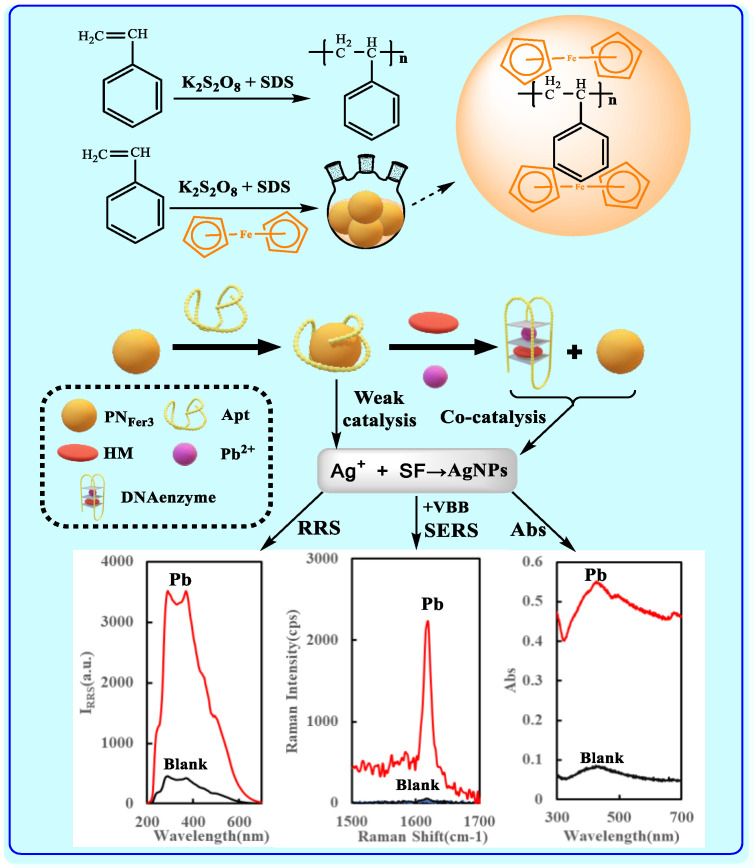
Preparation of ferrocene-doped polystyrene nanospheres and trimode detection of trace Pb^2+^ based on ferrocene-doped polystyrene and DNA enzyme cocatalysis of AgNO_3_-sodium formate nanosilver indicator reaction.

**Figure 2 nanomaterials-12-01243-f002:**
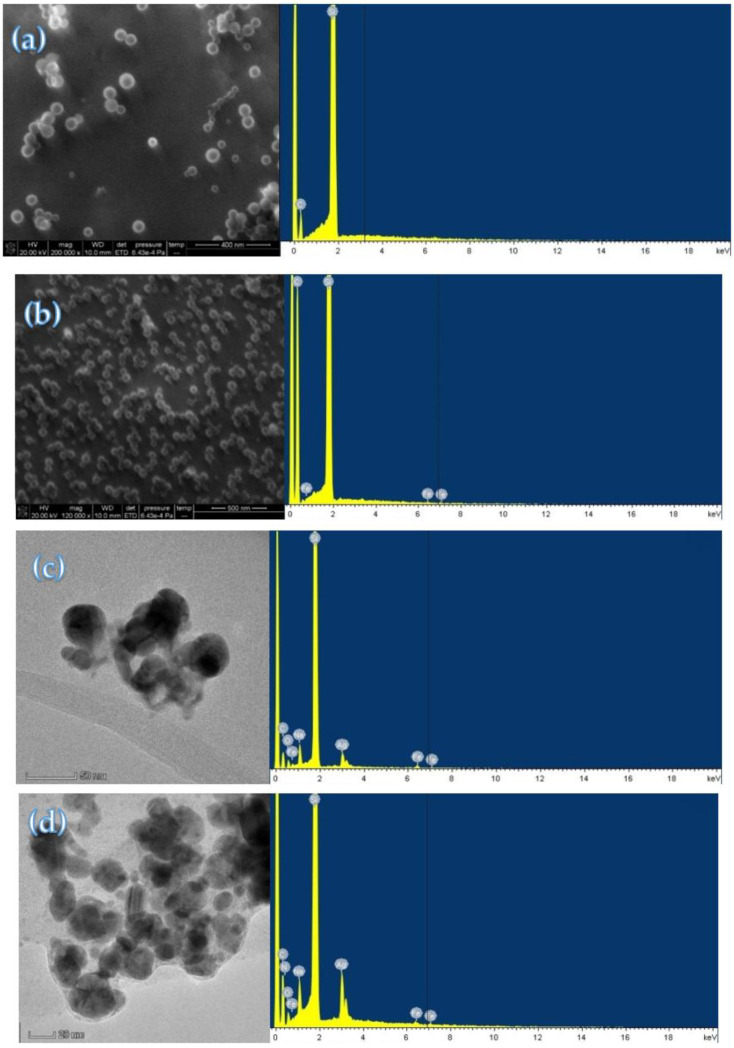
Electron microscope images and energy spectra of PN_Fer_ and nanocatalytic analysis system. (**a**) 6.67 mg/L PN_3_; (**b**) 6.67 mg/L PN_Fer3_; (**c**) 6 nM HM + 8 nM Apt + 6.67 mg/L PN_Fer3_ + 0.67 mM AgNO_3_ + 53.2 mM SF; (**d**) 1.0 nM Pb^2+^ + 6 nM HM + 8 nM Apt + 6.67 mg/L PN_Fer3_ + 0.67 mM AgNO_3_ + 53.2 mM SF.

**Figure 3 nanomaterials-12-01243-f003:**
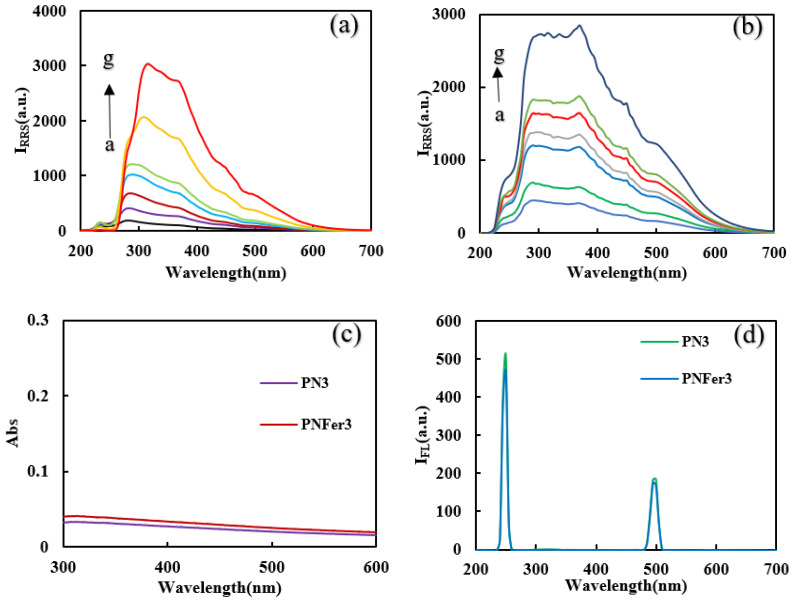
RRS, Abs and fluorescence spectra of polystyrene nanospheres. (**a**) RRS spectra of the curves a to g: (0.5, 1, 2, 3, 5, 10, 20) mg/L PN_3_; (**b**) RRS spectra of the curves a to g: (0.5, 1, 2, 3, 5, 10, 20) mg/L PN_Fer3_; (**c**) Abs spectra of 10 mg/L PN_3_ and PN_Fer3_; (**d**) fluorescence spectra of 10 mg/L PN_3_ and PN_Fer3_.

**Figure 4 nanomaterials-12-01243-f004:**
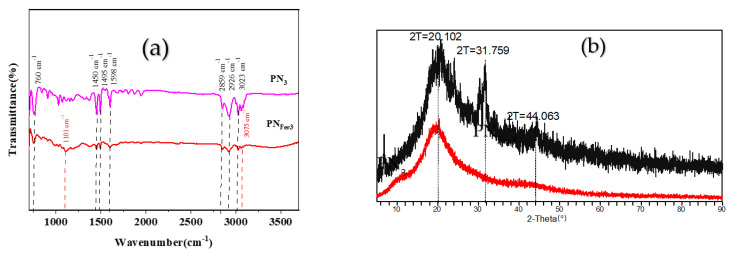
Infrared spectra and XRD pattern of polystyrene nanospheres. (**a**) Infrared spectra of PN_3_ and PN_Fer3_; (**b**) XRD of PN_3_ and PN_Fer3_.

**Figure 5 nanomaterials-12-01243-f005:**
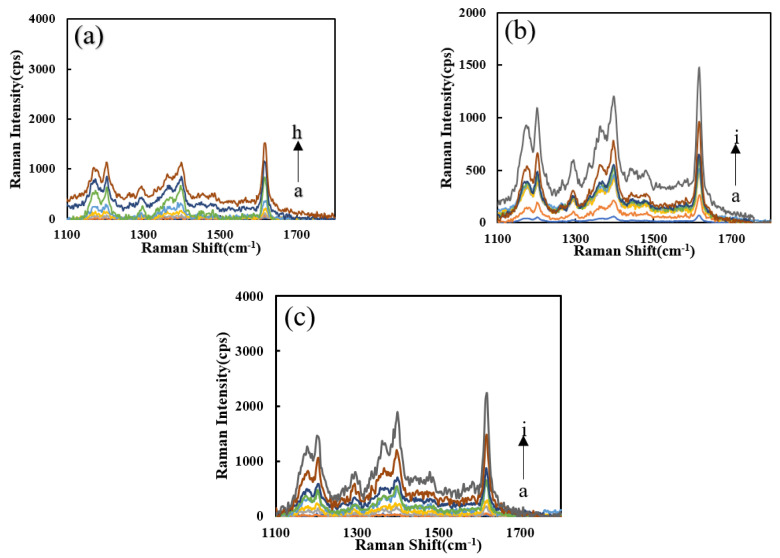
SERS spectra of the analysis system. (**a**) a to h: (0, 0.067, 0.13, 0.2, 0.27, 0.33, 0.67, 1) nM Pb^2+^ + HM + Apt + PN_3_ + AgNO_3_ + SF + VBB; (**b**) a to i: (0, 0.05, 0.1, 0.2, 0.3, 0.4, 0.5, 1, 2) nM Pb^2+^ + PN_3_ + Apt + HM + AgNO_3_ + SF; (**c**) a to i: (0, 0.05, 0.1, 0.2, 0.3, 0.4, 0.5, 1, 2) nM Pb^2+^ + PN_Fer3_ + Apt + HM + AgNO_3_ + SF.

**Figure 6 nanomaterials-12-01243-f006:**
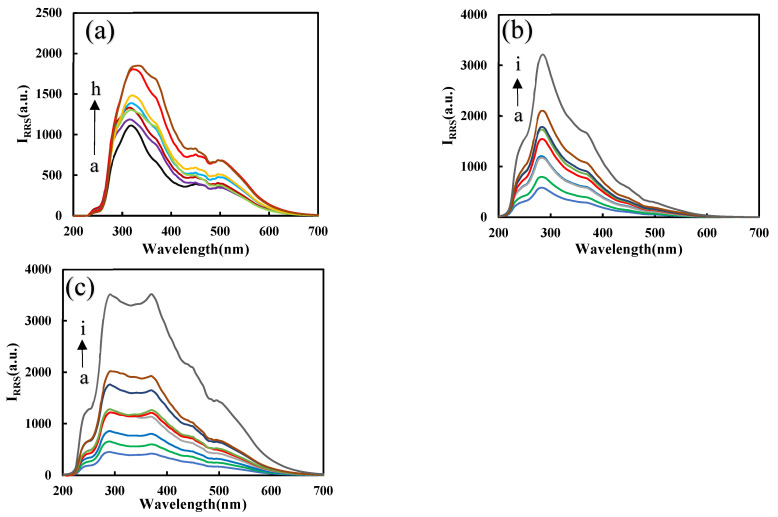
RRS spectra of the analysis system. (**a**) a to h: (0, 0.067, 0.13, 0.2, 0.27, 0.33, 0.67, 1) nM Pb^2+^ + Apt + PN_3_ + AgNO_3_ + SF. (**b**) a to i: (0, 0.05, 0.1, 0.2, 0.3, 0.4, 0.5, 1, 2) nM Pb^2+^ + PN_3_ + Apt + HM + AgNO_3_ + SF; (**c**) a to i: (0, 0.05, 0.1, 0.2, 0.3, 0.4, 0.5, 1, 2) nM Pb^2+^ + PN_Fer3_ + Apt + HM + AgNO_3_ + SF.

**Figure 7 nanomaterials-12-01243-f007:**
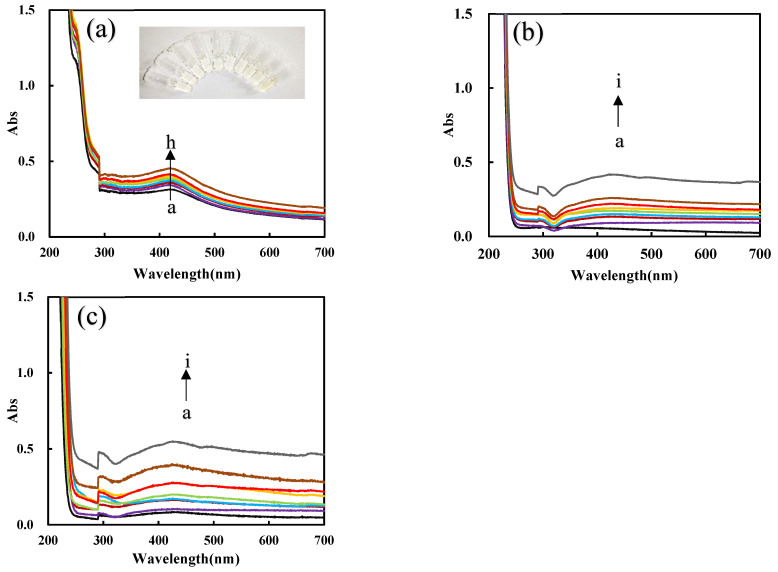
UV absorption spectra of the analysis system. (**a**) a to h: (0, 0.067, 0.13, 0.2, 0.27, 0.33, 0.67, 1) nM Pb^2+^ + HM + Apt + PN_3_ + AgNO_3_ + SF; (**b**) a to i: (0, 0.05, 0.1, 0.2, 0.3, 0.4, 0.5, 1, 2) nM Pb^2+^ + PN_3_ + Apt + HM + AgNO_3_ + SF; (**c**) a to i: (0, 0.05, 0.1, 0.2, 0.3, 0.4, 0.5, 1, 2) nM Pb^2+^ + PN_Fer3_ + Apt + HM + AgNO_3_ + SF.

**Figure 8 nanomaterials-12-01243-f008:**
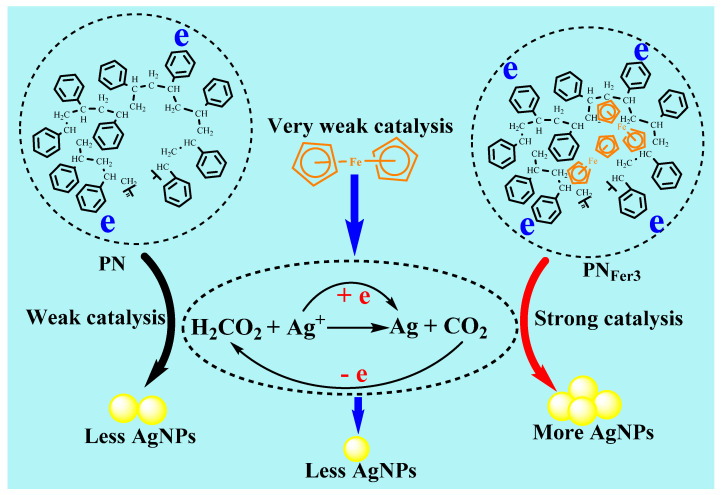
Comparison of polystyrene nanosphere (PN), Fer and ferrocene-doped polystyrene nanosphere (PN_Fer3_) nanocatalysis of AgNO_3_-SF reaction to form AgNPs.

**Table 1 nanomaterials-12-01243-t001:** Analytical characteristics of Pb^2+^ determination with different nanozymes.

Material	Method	Linear Range(nM)	Linear Equation	Coefficient	DL (nM)
PN_3_	RRS	0.1–1	ΔI = 742.6C_Pb_ + 155.2	0.835	0.06
	Abs	0.2–2	ΔA = 0.16C_Pb_ + 0.05	0.9381	0.1
	SERS	0.08–2	ΔI = 594.7C_Pb_ + 228.6	0.9307	0.03
PN_Fer3_	RRS	0.1–2	ΔI = 1432C_Pb_ + 246.8	0.9672	0.5
	Abs	0.2–2	ΔA = 0.23C_Pb_ + 0.04	0.9445	0.1
	SERS	0.05–2	ΔI= 1135C_Pb_ + 102.1	0.9651	0.03

**Table 2 nanomaterials-12-01243-t002:** Analytical results of Pb^2+^ in water samples.

Sample	Sigle Value (nM)	Average (nM)	Added (nM)	Found (nM)	Recovery (%)	RSD (%)	Content (μg/L)	Ref. Results (μg/L)
Farmland water 1	0.95, 0.94, 1.03, 1.02, 1.05	1.0	0.80	1.73	91.2	5.0	10.41	10.96
Farmland water 2	0.97, 0.97, 0.92, 1.01, 0.89	0.95	0.80	1.70	93.7	4.9	9.55	9.28
Lake water 1	0.92, 0.93, 0.85, 0.93, 0.92	0.91	0.80	1.78	108.7	3.7	9.41	9.86
Lake water 2	0.75, 0.82, 0.83, 0.78, 0.73	0.78	0.80	1.60	102.5	5.5	8.16	8.96
Waste-water	1.08, 1.17, 1.23, 1.03, 1.23	1.15	0.80	1.89	92.5	7.8	11.92	12.40

## Data Availability

Not applicable.

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
