# Peer review of "Ferrocene-Doped Polystyrene Nanoenzyme and DNAzyme Cocatalytic SERS Quantitative Assay of Ultratrace Pb2+"

_nanomaterials, 2022, doi:10.3390/nano12081243_

Round 1

Reviewer 1 Report

Recommendation: I do not recommend the manuscript “A Ferrocene-doped Polystyrene Nanoenzyme and DNAzyme Cocatalytic SERS Assay Platform for Pb2+” by  Chongning Li, Zhenghong Wang, Zhiliang Jiang for publication in the Nanomaterials journal.

The manuscript reports the synthesis of composite materials for which structure and composition are not correctly determined. Based on experiments presented in this work, it is not proved that the fabricated materials are catalytic active in reducing silver ions under the influence of SF. Such a reaction occurs at elevated temperature without a catalyst. Besides, the presence of ferrocene in the fabricated materials is questionable (the peak in EDS spectrum for Fe is on the noise level).

The manuscript is very poorly written, unclear, non-logical and full of substantive errors; below are a few examples:

“DNA enzymes (DNase) synthesized by aptamers have the advantages of specificity, simple synthesis, low price, and good stability. They have been used in metal ions, proteins, miRNA evaluation, bacterial detection and cell imaging”

“The results of energy spectrum show that there is peak at 0.46 keV ascribing to Fe element of Fer”

“Without of Pb2+, Apt inhibits the catalytic reaction, there are less spherical AgNPs in the system, with an average size of 42 nm.”

“Fig. S5(a) shows that with the increase of catalyzed Ag particles, the color of the so-lution gradually changes from light yellow to darker.”

“Due to its excellent electronic conductivity,29 ferrocene has a certain degree of thermal and chemical stability, but it is slightly soluble in water and difficult to disperse.”

“Accurately draw 10 mL of five water samples, first filtered with filter paper to remove suspended solids, then centrifuged at 12000 rpm/g”

“The water sample was measured 5 times, and then a certain amount of Pb2+ was added for the measurement of recovery.”

And many others…

Author Response

Recommendation: I do not recommend the manuscript “A Ferrocene-doped Polystyrene Nanoenzyme and DNAzyme Cocatalytic SERS Assay Platform for Pb2+” by Chongning Li, Zhenghong Wang, Zhiliang Jiang for publication in the Nanomaterials journal.

Answer: The manuscript was revised carefully according to the comments, such as the sections of Instrument and reagents, Preparation of polystyrene nanospheres, Preparation of ferrocene-doped polystyrene nanozyme, Characterization of PN3 and PNFer3 and Figure S3 SERS spectra of nanocatalysis and analysis system were moved to the text.

The manuscript reports the synthesis of composite materials for which structure and composition are not correctly determined. Based on experiments presented in this work, it is not proved that the fabricated materials are catalytic active in reducing silver ions under the influence of SF. Such a reaction occurs at elevated temperature without a catalyst. Besides, the presence of ferrocene in the fabricated materials is questionable (the peak in EDS spectrum for Fe is on the noise level).

Answer: The composite materials were characterized by the electron microscope and energy spectra, molecular spectra and XRD. It is a ferrocene-doped polystyrene nanosphere. The results show that the SF reduction of Ag(I) is slow to form nanosilver, the ferrocene-doped polystyrene nanosphere enhanced the nanoreaction. The aptamer inhibits the nanoreaction. Upon addition of Pb2+, the catalysis recovery.  The above were identified by the SERS, RRS and Abs results. The ferrocene-doped amount was small, the peak in EDS spectrum for Fe is weak.

The manuscript is very poorly written, unclear, non-logical and full of substantive errors; below are a few examples:

“DNA enzymes (DNase) synthesized by aptamers have the advantages of specificity, simple synthesis, low price, and good stability. They have been used in metal ions, proteins, miRNA evaluation, bacterial detection and cell imaging”

Answer: It was changed to “DNA enzymes (DNase) were synthesized simply by aptamers, with advantages of specificity, low price and good stability. It has been used in analytical chemistry, such as metal ions and proteins detection”.

“The results of energy spectrum show that there is peak at 0.46 keV ascribing to Fe element of Fer”

Answer: It was changed to “The energy spectrum (Fig.2b) shows that there are three peaks at 0.46, 6.4 and 7.1 keV, ascribing to Fe element of Fer”.

“Without of Pb2+, Apt inhibits the catalytic reaction, there are less spherical AgNPs in the system, with an average size of 42 nm.”

Answer: It was changed to “There are less spherical AgNPs in the blank system due to the inhibition of Apt, and the average size is 42 nm (Fig.2c).”

“Fig. S5(a) shows that with the increase of catalyzed Ag particles, the color of the so-lution gradually changes from light yellow to darker.”

Answer: It was changed to “Fig. S5(a) shows that the absorption peak at 430 nm increase with the PN3 catalyst concentration increasing, and the yellow gradually deepened.”

“Due to its excellent electronic conductivity,29 ferrocene has a certain degree of thermal and chemical stability, but it is slightly soluble in water and difficult to disperse.”

Answer: It was changed to “Fer is rich in electrons and has good thermal and chemical stability 29. Due to poor water solubility, it is rarely used in aqueous solution.”

“Accurately draw 10 mL of five water samples, first filtered with filter paper to remove suspended solids, then centrifuged at 12000 rpm/g”

Answer: It was changed to “A 10 mL water samples were filtered with filter paper to remove suspended solids, then it was centrifuged at 12000 rpm/g”

“The water sample was measured 5 times, and then a certain amount of Pb2+ was added for the measurement of recovery.”

Answer: It was changed to “Each sample was determined in parallel for 5 times, and then a certain amount of Pb2+ was added in the sample to obtain the recovery.”

 And many others…

Answer: Many others were also revised.

Reviewer 2 Report

The manuscript reports results of an experimental study on the synthesis of a nanosol of nano-microspheres of ferrocene-doped polystyrene and on the use of the nanosol to detect ultratrace amounts of Pb2+.

The manuscript claims that the nanosol has excellent catalytic ability for the indicator reaction of AgNO3-sodium formate to generate nanosilver with strong trimode spectral signals.  Additionally, the manuscript claims that the method allows for the detection of As3+ and Hg2+.

The manuscript reports interesting work, worth reporting in nanomaterials.  However, in its present form, the manuscript suffers in terms of both content and from.  In terms of content:

  1. The most upsetting issue is that a great deal of the meaningful experimental data is pushed into the supplementary materials, which makes the manuscript difficult to read and fractures the line of reasoning. I think that since length is not a major issue for an online article, the manuscript would gain significantly if the data are thoroughly described and interpreted in the paper.  I strongly suggest that most of the data is shown and properly discussed in the actual paper.
  2. The Introduction could gain in clarity if it started from the relevance of the study, continued with the state of the art of Pb2+ detection and the existing obstacles and limitations, to build the understanding of the choices made in the present work. Instead, the manuscript starts from some material choices that are poorly explained.  The state of the art in the field is insufficiently described, and the open questions are not formulated with enough clarity.  Some of the drawbacks in detecting Pb2+ are presented but the reasoning behind the new ideas proposed in the paper is not obvious.  The reader has to wait until the last sentence of the Introduction to begin to understand what the paper is actually reporting.  I recommend that the Introduction is carefully rewritten, to emphasize the existing limitations and to better highlight the novelty of the present work.
  3. Section 3 starts rather abruptly with the principle of trimode detection of trace Pb2+, without presenting evidence for the results of the synthetic effort. Those data and their discussions are ‘hidden’ in the supplementary materials. 
  4. The many curves in figures S3, S4 and S5 are insufficiently described and the captions extremely difficult to read. The numerous abbreviations make the compounds difficult to identify.  Given the complexity of materials and notations, I recommend that all samples are explained in a separate table.  Consequently, the explanation of the observed behavior is modest and the reasoning behind the results of the optimal sample remains insufficiently described.
  5. The claim in the abstract and in the manuscript that the method allows for the detection of As3+ and Hg2+ is not substantiated.

In terms of form:

  1. The use of acronyms is excessive, making the paper difficult to read.
  2. The figure captions provide insufficient information and are not self-explanatory. For instance, Fig. 1 is complex, with several boxes, but the caption fail to describe the content.  The caption of Fig. 2 is even less descriptive and the entries in Table 1 could benefit from a more detailed explanation.
  3. The use of English is awkward throughout the text. Leaving aside the numerous grammar mistakes, of concern is the convoluted way in which the sentences are constructed, hindering comprehension.  An example that strikes your eye is the “very less” phrase, which is bold in Fig. 2 and begs for a correction.

Under these circumstances, although the manuscript reports interesting results, it needs a major revision before it can be accepted for publication.

Author Response

The manuscript reports results of an experimental study on the synthesis of a nanosol of nano-microspheres of ferrocene-doped polystyrene and on the use of the nanosol to detect ultratrace amounts of Pb2+.

The manuscript claims that the nanosol has excellent catalytic ability for the indicator reaction of AgNO3-sodium formate to generate nanosilver with strong trimode spectral signals.  Additionally, the manuscript claims that the method allows for the detection of As3+ and Hg2+.

The manuscript reports interesting work, worth reporting in nanomaterials.  However, in its present form, the manuscript suffers in terms of both content and from.  In terms of content:

  1. The most upsetting issue is that a great deal of the meaningful experimental data is pushed into the supplementary materials, which makes the manuscript difficult to read and fractures the line of reasoning. I think that since length is not a major issue for an online article, the manuscript would gain significantly if the data are thoroughly described and interpreted in the paper.  I strongly suggest that most of the data is shown and properly discussed in the actual paper.

Answer: The sections of Instrument and reagents, Preparation of polystyrene nanospheres, Preparation of ferrocene-doped polystyrene nanozyme, Characterization of PN3 and PNFer3 and Figure S3. SERS spectra of nanocatalysis and analysis system were moved to the text.

  1. The Introduction could gain in clarity if it started from the relevance of the study, continued with the state of the art of Pb2+ detection and the existing obstacles and limitations, to build the understanding of the choices made in the present work. Instead, the manuscript starts from some material choices that are poorly explained.  The state of the art in the field is insufficiently described, and the open questions are not formulated with enough clarity.  Some of the drawbacks in detecting Pb2+ are presented but the reasoning behind the new ideas proposed in the paper is not obvious.  The reader has to wait until the last sentence of the Introduction to begin to understand what the paper is actually reporting.  I recommend that the Introduction is carefully rewritten, to emphasize the existing limitations and to better highlight the novelty of the present work.

Answer: Thank you for your good! We revised the section of Introduction. The paragraph of Pb2+ detection was moved to the first paragraph, and the existing obstacles and limitations were added. The paragraph of DNAzyme was moved to the second paragraph. The serial number of references has also been adjusted.

  1. Section 3 starts rather abruptly with the principle of trimode detection of trace Pb2+, without presenting evidence for the results of the synthetic effort. Those data and their discussions are ‘hidden’ in the supplementary materials. 

Answer: The section of 3 was revised. The preparation of polystyrene nanospheres, Preparation of ferrocene-doped polystyrene nanozyme, Characterization of PN3 and PNFer3 and Figure S3 SERS spectra of nanocatalysis and analysis system were moved to the section.

  1. The many curves in figures S3, S4 and S5 are insufficiently described and the captions extremely difficult to read. The numerous abbreviations make the compounds difficult to identify.  Given the complexity of materials and notations, I recommend that all samples are explained in a separate table.  Consequently, the explanation of the observed behavior is modest and the reasoning behind the results of the optimal sample remains insufficiently described.

Answer: About the explanation and captions of figures 2, Figure S1 and S2 were revised in the 3.3 and SM. All abbreviations appear for the first time in the text.

  1. The claim in the abstract and in the manuscript that the method allows for the detection of As3+ and Hg2+ is not substantiated.

Answer: The detection of As3+ and Hg2+ was deleted.

In terms of form:

  1. The use of acronyms is excessive, making the paper difficult to read.

Answer: In the text description, we try to reduce abbreviations as much as possible.

  1. The figure captions provide insufficient information and are not self-explanatory. For instance, Fig. 1 is complex, with several boxes, but the caption fail to describe the content.  The caption of Fig. 2 is even less descriptive and the entries in Table 1 could benefit from a more detailed explanation.

Answer: The figure 1, the figure captions and Table 1 were revised.

  1. The use of English is awkward throughout the text. Leaving aside the numerous grammar mistakes, of concern is the convoluted way in which the sentences are constructed, hindering comprehension.  An example that strikes your eye is the “very less” phrase, which is bold in Fig. 2 and begs for a correction.

Answer: The English was revised.

Under these circumstances, although the manuscript reports interesting results, it needs a major revision before it can be accepted for publication.

Answer: The manuscript was revised carefully according to the comments.

Reviewer 3 Report

The topic of the manuscript presented by Li et al. seems to be interesting. However, it is very difficult to read the paper and to follow the investigation line, to understand the discussed data. First of all, the English of the paper must be significantly improved. Next, from the introduction part, it is not clear at all what kind of experiments were done/data will be discussed in the manuscript. Why the authors did move the whole experimental part to the SM part? The basic and clear information of the materials and methods used in the work should be presented directly in the manuscript. Besides, the text of the SM is too dense, it is not clear at all, it is difficult to read and follow it. Please, take also care of the organization of the text and figures in the SM part.

The manuscript contains just 2 schematic diagrams and the whole discussion is done based on the results presented in the SM part. Why? It is very confusing that you are discussing something that cannot be found in the manuscript. Yes, there are a lot of plots. However, there is no need to present them all in the manuscript. Nevertheless, there is a way how to present to obtained data in a simple form. But first of all, it must be clear what you are analyzing, discussing, comparing, etc. For that, you can present in the manuscript a representative spectrum (data) and complete it with a comparative graph where the discussed parameter(s) will be depicted for all studied systems. By the way, there is no clear definition of the systems which you are discussing, so it is a little messy when suddenly the reader sees something like “HM-PB2+-Apt-PN3-AgNO3-SF-VBB system”. Please, explain all the used abbreviations before their first use. The legends must be written correctly (described completely and clearly). The writing/typing errors are very frequently found in the text. Please, check/correct them. The title(s) of the manuscript as well as paragraphs can be improved. For example, “2.2 Procedure”. What kind of procedure? There are also many sentences/parts which are not very clear and complete. For example, “The characteristics of PN and PNFer3 are also analyzed and detected (Fig. S2).” What kind of characteristics? Why are they important? OR “Changing the amount of styrene in the slow reaction process can cause polystyrene. The particle size of the microsphere changes. Experiments have found that when an appropriate amount of styrene was added, the microspheres prepared by the reaction have the best catalytic performance … The catalytic effect does not increase but decrease. …” It is not very clear. How it can be seen? What is the appropriate amount? Is it something that was measured/is discussed in the paper or has been already done? Maybe some reference is needed. OR “Polystyrene nano-microsphere nanosol…” Nano? Micro?

I cannot recommend the manuscript in the present form for its publication in Nanomaterials.

Author Response

The topic of the manuscript presented by Li et al. seems to be interesting. However, it is very difficult to read the paper and to follow the investigation line, to understand the discussed data. First of all, the English of the paper must be significantly improved. Next, from the introduction part, it is not clear at all what kind of experiments were done/data will be discussed in the manuscript. Why the authors did move the whole experimental part to the SM part? The basic and clear information of the materials and methods used in the work should be presented directly in the manuscript. Besides, the text of the SM is too dense, it is not clear at all, it is difficult to read and follow it. Please, take also care of the organization of the text and figures in the SM part.

Answer: The manuscript was revised carefully. The Introduction, SM, Figures and text were revised. The materials and methods were moved to the text.

The manuscript contains just 2 schematic diagrams and the whole discussion is done based on the results presented in the SM part. Why? It is very confusing that you are discussing something that cannot be found in the manuscript. Yes, there are a lot of plots. However, there is no need to present them all in the manuscript. Nevertheless, there is a way how to present to obtained data in a simple form. But first of all, it must be clear what you are analyzing, discussing, comparing, etc. For that, you can present in the manuscript a representative spectrum (data) and complete it with a comparative graph where the discussed parameter(s) will be depicted for all studied systems. By the way, there is no clear definition of the systems which you are discussing, so it is a little messy when suddenly the reader sees something like “HM-PB2+-Apt-PN3-AgNO3-SF-VBB system”. Please, explain all the used abbreviations before their first use. The legends must be written correctly (described completely and clearly). The writing/typing errors are very frequently found in the text. Please, check/correct them. The title(s) of the manuscript as well as paragraphs can be improved. For example, “2.2 Procedure”. What kind of procedure? There are also many sentences/parts which are not very clear and complete. For example, “The characteristics of PN and PNFer3 are also analyzed and detected (Fig. S2).” What kind of characteristics? Why are they important? OR “Changing the amount of styrene in the slow reaction process can cause polystyrene. The particle size of the microsphere changes. Experiments have found that when an appropriate amount of styrene was added, the microspheres prepared by the reaction have the best catalytic performance … The catalytic effect does not increase but decrease. …” It is not very clear. How it can be seen? What is the appropriate amount? Is it something that was measured/is discussed in the paper or has been already done? Maybe some reference is needed. OR “Polystyrene nano-microsphere nanosol…” Nano? Micro?

Answer: The important results of Figure 2-4 were moved to the txt. The sections of Introduction, Results and discussion were revised. The “HM-PB2+-Apt-PN3-AgNO3-SF-VBB” was changed to “Pb2+-Apt- heme-PN3-AgNO3- formate-VBB”. All abbreviations were explained at first use. The legends were written. The writing/typing errors were corrected. The title was changed to “Ferrocene-Doped Polystyrene Nanoenzyme and DNAzyme Cocatalytic SERS quantitative Assay of Ultratrace Pb2+’. The “2.2 Procedure” was changed to “2.2 Procedure for RRS and SERS detection of Pb2+”. The nanosphere was used.

I cannot recommend the manuscript in the present form for its publication in Nanomaterials.

Answer: The manuscript was revised carefully according to the comments.

Round 2

Reviewer 1 Report

I do not recommend the manuscript “A Ferrocene-doped Polystyrene Nanoenzyme and DNAzyme Cocatalytic SERS Assay Platform for Pb2+” by  Chongning Li, Zhenghong Wang, Zhiliang Jiang for publication in the Nanomaterials journal.

The revision prepared by Authors is not enough and the manuscript is still very poorly written, unclear, non-logical and full of substantive errors.

Author Response

I do not recommend the manuscript “A Ferrocene-doped Polystyrene Nanoenzyme and DNAzyme Cocatalytic SERS Assay Platform for Pb2+” by Chongning Li, Zhenghong Wang, Zhiliang Jiang for publication in the Nanomaterials journal.

The revision prepared by Authors is not enough and the manuscript is still very poorly written, unclear, non-logical and full of substantive errors.

Answer: The manuscript was revised by “Track Changes” function. The original Figure 3 was separated to the Figure 3 and 4. Some data of the original Figure 4 were moved to MS (Figs. S1), some important data of Figs. S1 and S2 about analytical system were moved to the Figure 6 and 7. The Table S4 was changed to Table 2 in the text. The above contents better support the topic. And the English was improved greatly.

Reviewer 2 Report

The revised manuscript has been significantly improved. In particular, the Introduction has clearly gained in clarity and description of the state of the art in the field has been improved.  Section 3 has been amended, presenting evidence for the results of the synthetic effort.  The captions for the figures and the tables have been extended and are more descriptive.  The claim in the abstract and in the manuscript that the method allows for the detection of As3+ and Hg2+, which was not substantiated in the text was removed.

As far as I can see, the paper has several key points.  The first message is the successful synthesis and characterization of the nanosol, which is now described with Figs. 1, 2 and 3.  The second point is that the nanosol has catalytic ability for the generation of nanosilver with strong surface-enhanced Raman scattering (SERS), resonance Rayleigh scattering (RRS) and surface plasmon resonance absorption (Abs).  The evidence for the trimode molecular spectral signals is shown in Figs. 3 and 4 in the manuscript and in Figs. S1 and S2 in the supplementary materials.  The third idea is that the nanocatalytic indicator could detect traces of Pb2+ in water samples.  The proof for that claim is shown in Table S4.

Under these circumstances, I think that some issues still remain and need to be addressed.  To improve clarity, I still suggest that the evidence is shown and properly discussed in the actual paper, leaving only minor data for the supplementary materials.  More specifically,

  1. To better support with evidence the first point, the results of the synthesis, Fig. 3 could be separated, to focus on the sample characterization.
  2. To prove the second point, that the nanocatalystt is a trimode indicator it may be preferable to bring as evidence the remaining data in Fig. 3, the most relevant data in from Fig. 4 and, similarly, the most relevant data from Figs. S1 and S2. The rest of the Fig. 4 and Figs. S1 and S2 could stay in the supplementary materials.  This way the main discussion stays in the text and would have correspondence in plots included the paper itself.  The supplementary materials remain only to show completeness of the data.
  3. To justify the third point, the detection of Pb2+, Table S4 should be included, in my opinion, in the paper. The inclusion could also serve a second purpose, to more clearly explain the samples studied.  The notations in the caption of Fig. 4 are still obscure and I continue to recommend that all samples are explained in a table and given a simple name/code that is used consistent in all figures to simplify captions and facilitate understanding.

In addition, the use of English is still awkward throughout the text.  The paper should be read by a proficient user. 

Under these circumstances, I think that the revised manuscript has been significantly improved, reports interesting results, but still needs a revision before it can be accepted for publication.

Author Response

The revised manuscript has been significantly improved. In particular, the Introduction has clearly gained in clarity and description of the state of the art in the field has been improved.  Section 3 has been amended, presenting evidence for the results of the synthetic effort.  The captions for the figures and the tables have been extended and are more descriptive.  The claim in the abstract and in the manuscript that the method allows for the detection of As3+ and Hg2+, which was not substantiated in the text was removed.

As far as I can see, the paper has several key points.  The first message is the successful synthesis and characterization of the nanosol, which is now described with Figs. 1, 2 and 3.  The second point is that the nanosol has catalytic ability for the generation of nanosilver with strong surface-enhanced Raman scattering (SERS), resonance Rayleigh scattering (RRS) and surface plasmon resonance absorption (Abs).  The evidence for the trimode molecular spectral signals is shown in Figs. 3 and 4 in the manuscript and in Figs. S1 and S2 in the supplementary materials.  The third idea is that the nanocatalytic indicator could detect traces of Pb2+ in water samples.  The proof for that claim is shown in Table S4.

Under these circumstances, I think that some issues still remain and need to be addressed.  To improve clarity, I still suggest that the evidence is shown and properly discussed in the actual paper, leaving only minor data for the supplementary materials.  More specifically,

  1. To better support with evidence the first point, the results of the synthesis, Fig. 3 could be separated, to focus on the sample characterization.

Answer: It was separated to the Figure 3. RRS, Abs and fluorescence spectra and the Figure 4. Infrared spectra and XRD pattern of polystyrene nanospheres. The section was revised.

To prove the second point, that the nanocatalystt is a trimode indicator it may be preferable to bring as evidence the remaining data in Fig. 3, the most relevant data in from Fig. 4 and, similarly, the most relevant data from Figs. S1 and S2. The rest of the Fig. 4 and Figs. S1 and S2 could stay in the supplementary materials.  This way the main discussion stays in the text and would have correspondence in plots included the paper itself.  The supplementary materials remain only to show completeness of the data. Answer: Thank you! Some data of the original Figure 4 were moved to MS (Figs. S1), some important data of Figs. S1 and S2 about analytical system were moved to the Figure 6 and 7.

To justify the third point, the detection of Pb2+, Table S4 should be included, in my opinion, in the paper. The inclusion could also serve a second purpose, to more clearly explain the samples studied.  The notations in the caption of Fig. 4 are still obscure and I continue to recommend that all samples are explained in a table and given a simple name/code that is used consistent in all figures to simplify captions and facilitate understanding.

Answer: The Table S4 was moved to the text as Table 2. The caption of the original Fig. 4, and the first column of Table 2 were revised. The section of 3.8 was revised.

In addition, the use of English is still awkward throughout the text.  The paper should be read by a proficient user. 

Answer: The English was revised.

Under these circumstances, I think that the revised manuscript has been significantly improved, reports interesting results, but still needs a revision before it can be accepted for publication.

Answer: Thank you! The manuscript was revised according to your comments.

Reviewer 3 Report

Dear authors,

I appreciated the changes you have made. However, most of them have been concerned about moving the information from SM to the main text of the manuscript. In any case, there were no changes regarding the improvement of the discussed figures as well as the discussion at all. It is still not so evident=straightforward that the fabricated materials are catalytic active in reducing silver ions under the influence of SF. Besides, I have to recommend once again to improve the English language of the manuscript. It is still written/read with high difficulty.

Author contribution: Who is C.N.? Why Z.H. instead of Z.W.?

In conclusion, an additional minor revision of the manuscript is highly recommended.

Author Response

I appreciated the changes you have made. However, most of them have been concerned about moving the information from SM to the main text of the manuscript. In any case, there were no changes regarding the improvement of the discussed figures as well as the discussion at all. It is still not so evident=straightforward that the fabricated materials are catalytic active in reducing silver ions under the influence of SF. Besides, I have to recommend once again to improve the English language of the manuscript. It is still written/read with high difficulty.

Answer: The nanoparticle catalysis of AgNO3-sodium formate was demonstrated by Figure S2(a). The English was revised.

Author contribution: Who is C.N.? Why Z.H. instead of Z.W.?

Answer: Thanks! The full name was replaced the abbreviation in the place of Author Contributions.

In conclusion, an additional minor revision of the manuscript is highly recommended.

Answer: The manuscript was revised carefully.